# "Pro-Raphaelites": The Classical Ideal in Religious Art and the Agency of Artworks in Estonia from 1810 to 1840

**Liisa-Helena Lumberg-Paramonova**

Institute of Art History and Visual Culture, Estonian Academy of Arts, 10412 Tallinn, Estonia;
liisahelena.lumberg@artun.ee

**Abstract:** This article analyzes Baltic German religious art based on examples from Estonia in the first half of the nineteenth century, focusing on artistic networks and the reclamation of a Renaissance classical ideal. Baltic German artists such as Friedrich Ludwig von Maydell, Gustav Adolf Hippius, and Otto Friedrich Ignatius were in contact with the Nazarenes, whose ideals were inspired by Raphael's attempt to merge art and religion. The Nazarenes influence can be seen in Baltic German religious art, which favored idealized forms and followed on from the works of the Renaissance masters. In addition to presenting religious scenes, in the Baltic context, these artworks acted as mediators of European artistic heritage. The classical ideal was thus perpetuated by a tightly connected network in which Baltic German artists joined others in re-establishing the power of the European canon of art history.

**Keywords:** Baltic Germans; Nazarenes; artistic contact; copies; Baltic art history; religious art; classical ideal; 19th century

## 1. Introduction

The cultural sphere in the early nineteenth-century Baltic Provinces (Estonia, Livonia, and, after 1795, Curonia) that formed the westernmost part of the Russian Empire was controlled by the Baltic Germans, who formed an upper-class minority amongst the Estonian and Latvian peasant majority. Written culture in the Baltic Provinces may largely be described as German colonial culture that oriented itself towards the German "motherland" and was thus connected with wider German-language cultural spheres. An important carrier of Baltic German culture was the scarce but gradually growing *Litteratenstand*—the university-educated bourgeoisie mostly consisting of pastors, doctors, teachers, and professors (Lukas 2021, p. 14).

Most of the intellectuals active in this region originated from Germany, mediating ideas and attitudes from there. The Baltic Provinces suffered damages in the Great Northern War (1700–1721) that hindered cultural life and education and highlighted the importance of immigrated teachers, pastors, artists, etc. In 1802, Tartu (Dorpat) University, the only German-speaking university in the Russian Empire, was re-opened. Many professors were invited from Germany, such as Johann Karl Simon Morgenstern (1770–1852), who established the university library and art museum and taught and specialized in aesthetics and art history (as well as other fields).

Nineteenth-century art has generally been addressed from a progress-oriented, Francophile viewpoint that omits a major part of its broader heritage. This analysis rejects the typical, center-periphery model in favor of variegated microhistories (Facos 2019, p. xxv). Religious art from the beginning of the nineteenth century as a separate topic has not been thoroughly researched in Estonia. A major study, Tiina-Mall Kreem's dissertation "Built as expected. Church building, church architecture and church art during the reign of Alexander II (1855–1881) in Estonia (Kreem 2010)", is focused on the art in churches during a later period.

Cordula Grewe's research has reinvigorated the study of religious art in the early nineteenth century by demonstrating the complex intellectual formation of aesthetic ideals in the realm of religious art of the Brotherhood of St Luke (*Lukasbund*), also called the Nazarenes. Grewe's work has transformed the perception of nineteenth-century Nazarene art—from retrograde to conceptually sophisticated (Grewe 2009, 2015). This new emphasis on Nazarene art is particularly important for Baltic art history because Friedrich Ludwig von Maydell, Gustav Adolf Hippius, and Otto Friedrich Ignatius had direct contact with Nazarene artists in Rome.

This article argues that the ties between Baltic German and Nazarene artists in the early nineteenth century mobilized an important yet heretofore neglected area of Baltic art history: debates about the classical ideal in religious art. In this paper, I analyze Baltic German religious art at the beginning of the nineteenth century in the context of different theories of religious art in the period. In order to understand these relationships, I explore the discourse of religious art in Germany and how it inspired Baltic German artists. Among the most important legacies of this artistic exchange were debates regarding the "classical ideal" in contemporary religious art.

The classical ideal is a key concept in European art and art historiography, establishing the style of a discrete period as the model for exemplary artistic practice. Since early modern treatises circulated classical norms through the printing press, the notion of the "classical ideal" has always depended on its mediation. Far from standardized, the classical ideal is actually contingent upon the agents of its dissemination (Gell 1998). Since each generation of artists redefines the classical ideal, it is at once fixed and mutable (Kodres 2003; Locher 2012; Brzyski 2007; Elkins 2007). Therefore, the classical ideal (or canon) does not exist per se, but presumes constant recreation and affirmation.

Debates surrounding the classical ideal directly impacted religious art in the first half of the nineteenth century because artists idealized transcendental ideas and aspects of the heavenly in order to distinguish them from the mundane. If mid-nineteenth-century British artists who diverged from this notion called themselves the Pre-Raphaelites, followers of Renaissance artists before Raphael, their Baltic German contemporaries might be best understood as "Pro-Raphaelites".

In her doctoral thesis, "Enlightenment Art. Baltic Dilettanti and Drawing Practices at the Turn of the Nineteenth Century" (2019), Estonian art historian Kadi Polli eschews the harsh distinctions between 'high' and 'low' genres in order to incorporate amateur drawing practice in research. The latter was of great importance in the Baltics, carrying Enlightenment ideas and disseminating knowledge about art. This demonstrates the utility of analyzing pictures as phenomena, focusing on their potential to influence and act (Polli 2019, p. 21), especially in an era of increased print-circulation (p. 52). Taking a cue from Polli's interventions, this article also investigates how ideas regarding religious art were put into practice in the aesthetic sphere and identifies what values were prioritized and which artistic means were chosen in order to represent them. The extent to which individual works were copied suggests that they had agency beyond their original settings. How, then, did these works act as mediators of ideas across the Baltic Sea?

It has to be kept in mind that the borders of the Baltic Provinces at the beginning of the 19th century differed greatly from the borders of the nation states today. Although the province of Livonia encompassed a large part of today's Estonia and Latvia, artworks mostly from Estonia are examined here in order to limit the scope of this article. Firstly, this article provides an overview of religious art in Estonia, focusing on copies of the works of the Renaissance masters or prominent German artists that functioned as mediators of European cultural heritage and ensured the clarity and unambiguousness of religious art. Secondly, this article explores the contact between Baltic German artists and the Nazarenes, briefly describing the travels of the aforementioned Hippius, Ignatius, and Pezold and their corresponding attitudes towards Nazarene art. Thirdly, some prominent examples of religious artworks are examined, analyzing both their form and content in relation to general discussions and tendencies regarding religious art.

## 2. Religious Art in Estonia in the First Half of the 19th Century: Copies as Mediators of Cultural Heritage

In present-day Estonia, religious art is housed in either museum collections or in churches. These collections largely consist of the portraits and landscapes that were most popular at the beginning of the nineteenth century. Religious scenes appear as prints more often than as paintings, which is natural, considering the lower price and reproducibility of prints. There were virtually no artists in the Baltic Provinces, Baltic German or otherwise, who specialized exclusively in religious art.[1]

While the most prolific artist of the first half of the century, Carl Sigismund Walther, was primarily a portraitist, he also created altar paintings for some 20 Lutheran churches in Estonia. His original compositions followed established conventions. Most of his altar pieces depicted Christ on the Cross, with two or more additional figures arranged in a pyramidal grouping. Fellow Baltic German painters Wilhelm von Kügelgen (*Christ on the Cross* on St. Olaf's Church altarpiece, 1833, Figure 1) and Ludwig von Maydell (*Christ at Gethsemane* in Saarde (Saara) Church, 1835; *Resurrection* in Põlva (Pölwe) Church, 1845) also created original compositions for altarpieces.

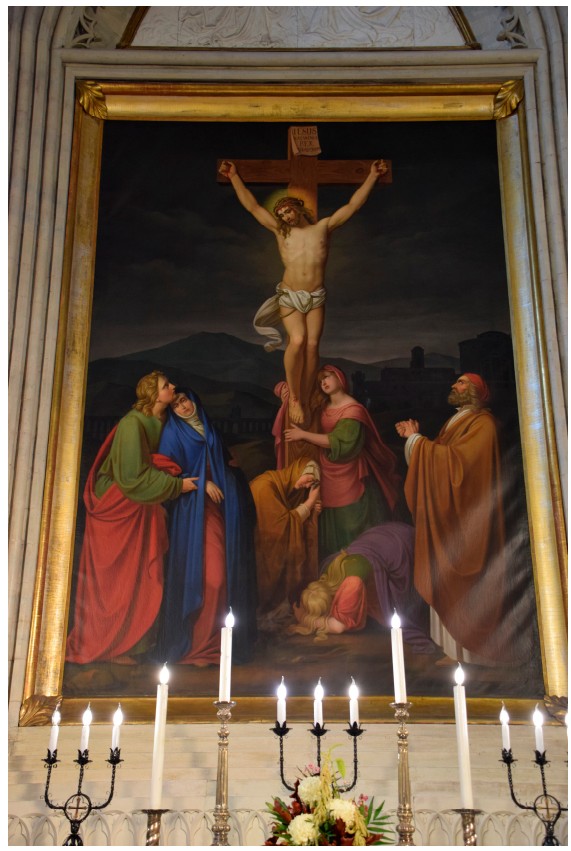

**Figure 1.** Wilhelm Georg Alexander von Kügelgen. *Christ on the Cross*. Altar Painting in Saint Olaf's Church, Tallinn, Estonia. 1831–1833. Oil on canvas. 400 × 275 cm. Photo by the author.

A considerable proportion of altar paintings in the nineteenth century were copied from the work of either German artists, the famous Renaissance masters, or other local artists (Kreem 2007). Copies of the works of Henry James Richter (*Help me, Lord!*), an English illustrator with German origins, and Heinrich Ferdinand Hofmann (*Christ at Gethsemane*) were especially popular; there are about ten copies of works of each of these artists in Estonian Lutheran churches. Leonardo da Vinci's *Last Supper* was also popular among copyists and can sometimes be found elsewhere in churches, not only in the altarpiece. There also exist works copied from Raphael. One of the most prominent of these was by Ernst Gotthilf Bosse, who created the altar painting for the Riga Dome Church in 1817,

a copy of Raphael's *Transfiguration* that was admired for its exactness and closeness to the original (Der Maler Ernst Bosse in Rom 1820). Bosse visited Italy several times and maintained contact with German artists staying in Rome. He was vastly influenced by the Italian Renaissance masters (Scheunchen n.d.). A painting in Kodavere (Koddafer) church from about 1800 by an unknown artist is partly a copy of Raphael's *Transfiguration*. In the 1770s, a copy of Raphael's *Transfiguration* became the altar painting in Kirbla (Kirrefer) church; this work was probably painted by Baron Clodt.[2] A copy of *Transfiguration* can also be found in Nõva (Newe) church (a rare paper vitrage) beside a copy of Antonio da Correggio's *Nativity*. There are some copies of works by Paolo Veronese and Fra Angelico as well.

Several factors determined the abundance of copies in church art. Because congregations were often poor—especially in rural areas—commissioning altar paintings was based on pragmatic considerations, with copies being more affordable than original compositions. Supply and demand were equally robust in the replica market, and altarpieces could be ordered from artists in Germany (Kristov 1994, p. 46). For example, the altarpiece *Let the Children Come to Me*, which can be found in the Esku chapel, was purchased in Munich by the artist Johann Carl Koch in 1845 (27929 Altar painting *Let the Children Come to Me* n.d.). Originality was not considered the most important characteristic of an altar painting; more significant was compatibility with the theological views and liturgy that an altar painting had to support. It is interesting that Lutheran pastors did not have a problem with images originating from a Catholic background since the cultural capital and the "enchantment of masterpieces" attached to such works were probably more important than possible doctrinal conflict (Kreem 2007, p. 55).

Ecclesiastical art needed to be clearly legible and unambiguous. At the consecration of altar paintings, the pastor often explained the meaning of the painting to the congregation.[3] It is probably for this reason that the art canonized in Estonian churches tended to represent common iconographies. Copies mediated art's historical value, recreating a canon of art. The most often copied paintings were generally famous works considered to be of high quality in terms of both subject matter and form. Furthermore, they were already familiar enough to be recognized by the educated majority and understood without further explanation.

Sometimes, copies themselves transformed many times, as they were often based on graphic reproductions, rather than on the original paintings. Copies were often pastiches of various source-works rather than exact reproductions of single paintings. For example, *Ecce homo* by Guido Reni, one of the 'stars' of Christian art, was used by the Baltic German artist Otto Friedrich Theodor von Moeller for the head of Christ on his altar paintings in churches in Harju-Jaani (St. Johannis) in 1872 and Cēsis (Wenden) in 1873 (Kreem 2007, p. 49). As early as 1818, Carl Sigismund Walther, the first lithographer in the Baltics, disseminated Reni's work in the form of a lithograph made in Tallinn (Reval) (Figure 2). Often, the line between an original composition and a copy became blurred, particularly when the motif had been repeated so frequently that it was difficult to determine where copying ended and interpretation began. This was also a consequence of the style of copying at the time, which did not always try to imitate the original exactly but rather interpret it creatively (Kreem 2005, p. 38).

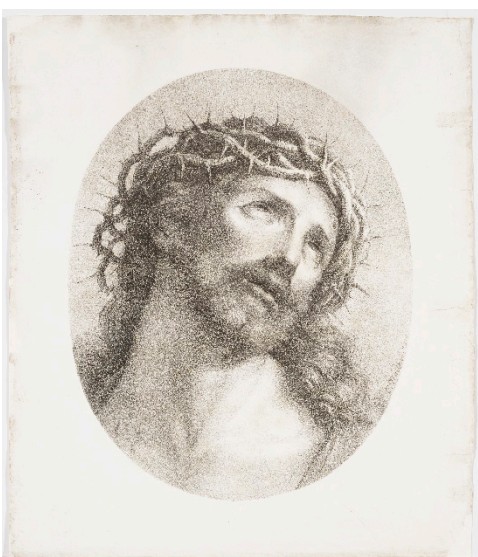

**Figure 2.** Carl Sigismund Walther. *Head of Christ with the Crown of Thorns*. 1818. Lithograph. 57 × 48.8 cm. Art Museum of Estonia, Tallinn. EKM j 57,112 G 30120.

### 3. Visiting the Brotherhood of St Luke: Baltic German Artists' Contact with the Nazarenes and the Reception of Raphael in Estonia

In 1809, six young artists formed the Brotherhood of Saint Luke and inhabited the monastery of Saint Isidore in Rome. Their ideas about religious art spread quickly and, by the middle of the nineteenth century, were influential throughout Europe (Grewe 2005, p. 43). Coming from a Romantic background, in objection to the rationalist ideas of the Enlightenment, the Nazarenes sought to re-mystify and aestheticize their surroundings. They found religion and art to be effective means for achieving this because both embodied transcendence; art enabled the attainment of the divine (ibid., p. 44.).

Nazarene art was conceptual; it sought to depict ideas. For them, an artwork was primarily a vessel for religious content (Grewe 2009, p. 307). This did not mean, however, that emotional content was ignored. A picture was meant to act in two ways: firstly, to influence feelings and induce piety, and secondly, to provide food for thought, thus moving from subjective influence to re-establishing common ideals (ibid., p. 310). To achieve this, the Nazarenes used generalized figures, focused on drawing rather than color, and gave their works a polished finish reminiscent of the Renaissance masters. The Nazarenes considered the harmonious symbiosis of emotion, intellectuality, and true religiosity to be embodied in the art of the Middle Ages and Raphael. Raphael was both seen as a paradigm after which to design one's life and artistic principles (Heß et al. 2012, p. ix) and venerated as a saint: the Nazarenes organized a pilgrimage to Urbino (Grewe 2012, p. 268), and when Raphael's remains were exhumed in 1833, they were honored as relics (Nerlich 2012, p. 49).

Leopold Pezold described the contacts of Baltic German artists with the Nazarenes in articles published in *Baltische Monatsschrift* between 1889 and 1890. Based on sketchbooks, diaries, letters, poems, oral accounts, and memories, he described the travels of his father, August Georg Wilhelm Pezold, and Gustav Adolf Hippius and Otto Friedrich Ignatius, who spent the years 1812–1819 studying and travelling in Europe (Pezold 1889, 1890a, 1890b, 1994). They met numerous artists along the way but were especially impressed by the German artists they met in Rome, the Nazarene Johann Friedrich Overbeck, in particular. Hippius was very fond of Overbeck's art, and they drew portraits of each other (Pezold 1890a, p. 35).[4]

The three travelers were also linked to the Nazarene circle by family ties. In Berlin, Ignatius and Pezold stayed at the home of sculptor Johann Gottfried Schadow. Schadow's two sons—Rudolph, a sculptor (like his father), and Friedrich Wilhelm, a Nazarene painter—were artists in Rome. Ignatius later married Adelheid, the daughter of Johann Gottfried

Schadow, and established warm friendships with her brothers (Pezold 1889, p. 720). Direct Nazarene influence is evident in religious art in Estonia as well. August Pezold, who painted *The Last Supper* for Türi (Turgell) Church, borrowed the composition from Johann Friedrich Overbeck, a fact that was mentioned as early as 1856 in the journal *Das Inland* (E. W. 1856).

Unlike the Nazarenes, Baltic German artists did not convert to Catholicism; instead, they remained Lutheran. What they gleaned from the Nazarenes was their visual language. At the same time, the celebrations of Catholic holidays and the aesthetics and sensitivity of Nazarene art were observed with interest and pleasure (Pezold 1890a, pp. 33–34). It is interesting to see how, for example, Hippius's views changed over time. Leopold Pezold mentions that Hippius was deterred by Nazarene Catholicism at first but gradually became more supportive of the Brotherhood (ibid., p. 37).

At this time, Raphael was one of the most popular artists to copy, an endeavor facilitated by graphic reproductions. Engravings of Raphael's works could be found in Estonian collections as early as the mid-seventeenth century (Hein 1993, p. 240). In the early nineteenth century, the newly re-established Tartu University bought many copies of Raphael's masterpieces for its art museum, and these were engraved by Giovanni Volpato, Raffaello Sanzio Morgheni, and others (Lääne-Euroopa Varane Graafika n.d.). Professor Karl Morgenstern played a key role in collecting art works for the University. His collection had over 1700 sheets, including numerous reproductions of Raphael's works (Kukk 1995, p. 10).

Morgenstern spread his knowledge in print as well, writing about Raphael in newspapers and journals, especially in his own publication, *Dörptische Beyträge für Freunde der Philosophie, Litteratur und Kunst* (Morgenstern 1813). He described his impressions from Paris, Dresden, and Saint Petersburg, where he had seen original works by Raphael. His writings were emotional, even ecstatic, especially regarding *The Sistine Madonna*, in which Raphael, according to Morgenstern, had demonstrated unsurpassed expressivity and superb composition, making all other paintings in Dresden's Gemäldegalerie a blurry haze (Morgenstern 1805, p. 15).

## 4. Religious Art in Practice: A Few Examples

During the three friends' travel in Europe, Ignatius seemed most interested in religious art, copying large compositions in Rome (Pezold 1890b, p. 111). He was later commissioned to paint the ceiling of the royal chapel in Tsarskoye Selo, near St. Petersburg. Although the chapel was destroyed in the World War II, a draft of Ignatius's design belongs to the Art Museum of Estonia (Figure 3). Like Raphael's *Disputa* in the Vatican, the painting is divided into two realms—celestial and earthly. The Virgin Mary, holding the Infant Jesus, sits in the clouds surrounded by angels and a halo of bright yellow light. Below, in the earthly sphere, are three female figures: Faith, holding a cross and a cup, Hope with an anchor, and bare-breasted Love (surrounded by four small children). On the pale blue horizon appears a mountain range conveyed by a Leonard-esque atmospheric perspective. On the left side, beside Faith, is a temple facade. On the right side, behind Hope, are rocks and a palm tree. Putti gaze down on the scene from the uppermost part of the composition.

Ignatius's study shows a complex, well-conceived, and typically Nazarene program. The figures are harmonious and idealized in a Raphael-esque manner; the colors are mild. The study shows rather Catholic and classical imagery, focusing on the Virgin Mary and the three theological virtues—Faith, Hope, and Love—whereas many nineteenth-century altar paintings in the Baltic Provinces depicted Christ on the Cross (Lumberg 2017, p. 67). The temple facade and Italian landscape complement this approach.

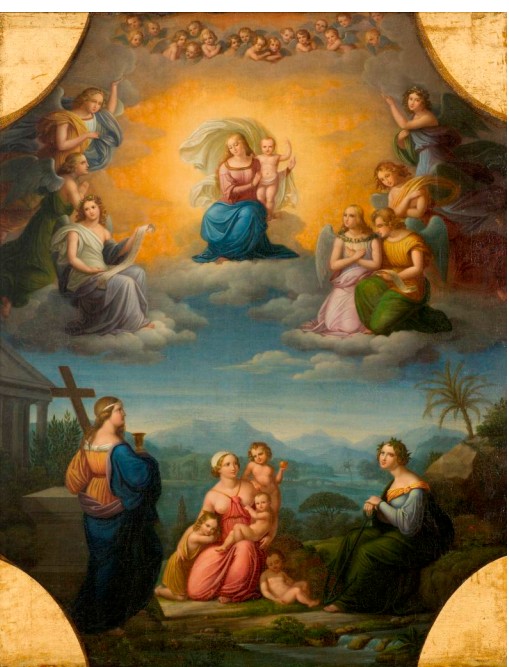

**Figure 3.** Otto Friedrich Ignatius. *Religious Scene.* Draft. 1822. Oil on canvas. 121.6 × 94.4 cm. Art Museum of Estonia, Tallinn. EKM j 6034 M 3038.

Since Ignatius died in 1824 at age 30, the ceiling painting of Tsarskoye Selo was completed by Hippius under the guidance of Carl Sigismund Walther. According to Pezold, Tsar Alexander I liked the painting very much (Pezold 1994, p. 94). Later in the nineteenth century, this work was copied for the altar painting in Hiitola Church, in Karelia,[5] a gift of Julius Ekmann. Unfortunately, the copyist and exact date are unknown (Hanka and Hätönen 1997, p. 289), but it nevertheless reveals the spread and acceptance of Ignatius's image and demonstrates how Baltic German contact with Nazarenes transformed religious art for new audiences throughout the multiconfessional Russian Empire.

Another important commission occurred under the auspices of the restoration of Tallinn's St. Olaf Church, which was damaged by fire in 1820. The stone altar wall, extraordinary in the Estonian context because of its material and scale, was designed by Ludwig von Maydell, who was also in contact with the Nazarenes during his time in Italy (Vaga 1976, p. 79). Both the neo-Gothic retable and the gilded bronze reliefs on the altar table—reminiscent of Ghiberti—were executed according to Maydell's drafts. The front panel of the altarpiece depicts *The Last Supper* (Figure 4), while the side-panels show the *Annunciation* and *The Baptism of Christ*.

A new altarpiece depicting *Christ on the Cross* (Figure 1) was commissioned from Wilhelm von Kügelgen, son of the popular (Baltic) German portraitist Franz Gerhard von Kügelgen.[6] Christ is in the center, towering above a group of mourners. This painting is also divided into celestial and earthly spheres, separated from each other by the zigzag silhouette of mountains in the background. The upper part of the picture is filled with a dark sky, which refers to the Biblical passage that states that, in the sixth hour, the sky darkened (Matthew 27:45). The lucid body of Christ is in strong contrast to the dark sky. In the lower part of the painting stands St John the Evangelist, supporting the Virgin Mary, with Mary Magdalene and other mourning ladies, as well as Joseph of Arimathea.

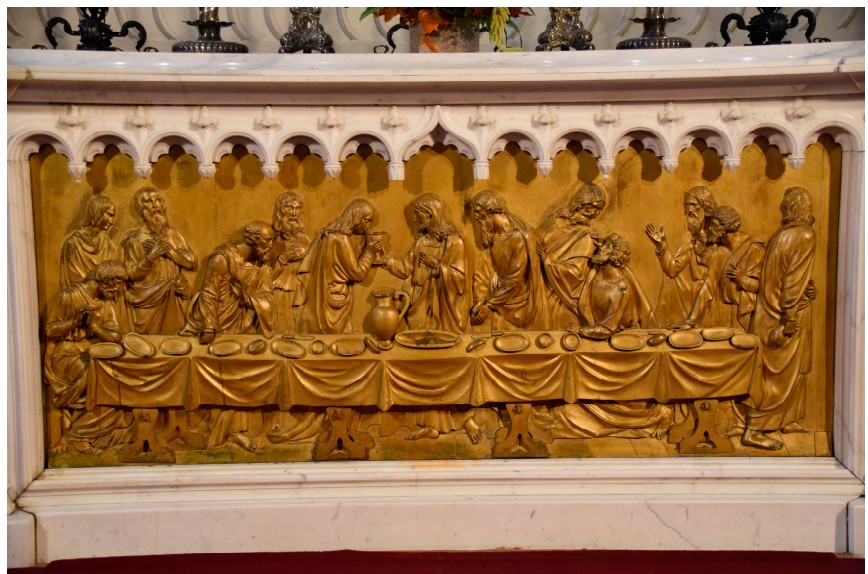

**Figure 4.** Friedrich Ludwig von Maydell (cast in bronze by Pavel Bazhenov). *Last Supper*. Detail from the Altarpiece in Saint Olaf 's Church, Tallinn, Estonia. 1830–1832. Gilded bronze. 78 × 190 cm. Photo by the author.

Here, too, the characters in the picture are dressed in robes that fall in folds. The buildings in the background—an aqueduct, a campanile, and a fortress—add an Italian quality. Although neither the clothing nor facial expressions are naturalistic, the artist sought to affect the viewer emotionally, creating a devotional image. The figures strike expressive poses, and their watery eyes complement their emotional facial expressions. From afar, it seems as if Christ looks tenderly at John and Mary, but upon closer inspection, one can see that Christ's eyes are already glassy—his intense suffering has already left him dying.

Kügelgen sought to combine emotion, intellectuality, and true religiosity, as was habitual to the Nazarenes. Concrete forms and figures acted as vessels for religious content, and in order to achieve that, a generalized form was preferable. Unlike writing, a visual artist had to specify details such as architecture or clothing, thus having to decide whether to set the scene in the Biblical times, the artist's own time, or some other time. How does one depict the heavenly and transcendental using particular, singular human means? This problem was mostly solved by choosing as generalized a depiction as possible, or by selecting a style/environment that was believed to carry universal values, such as Antiquity and the Renaissance, as its descendant.

Kügelgen also made religious prints (Figures 5 and 6). His illustrations of Old Testament stories were published in the series *The History of the Heavenly Kingdom According to Holy Scripture* (*Die Geschichte des Reichs Gottes, nach der heiligen Schrift*), accompanying texts by the German theologian Friedrich Adolf Krummacher (1833). In this series, a preference for linearity is evident—whole scenes are conveyed by outline alone. Borders filled with additional ornaments or figures to complement the main theme surround each scene, resembling *Morning* (1808), an influential painting by Philipp Otto Runge, who was born in Wolgast, studied in Copenhagen, and worked in Hamburg. This arrangement impedes the viewer's attention from wandering by transfixing it upon ideas inspired by the Bible, illustrated in the image, and expanded in the frame.

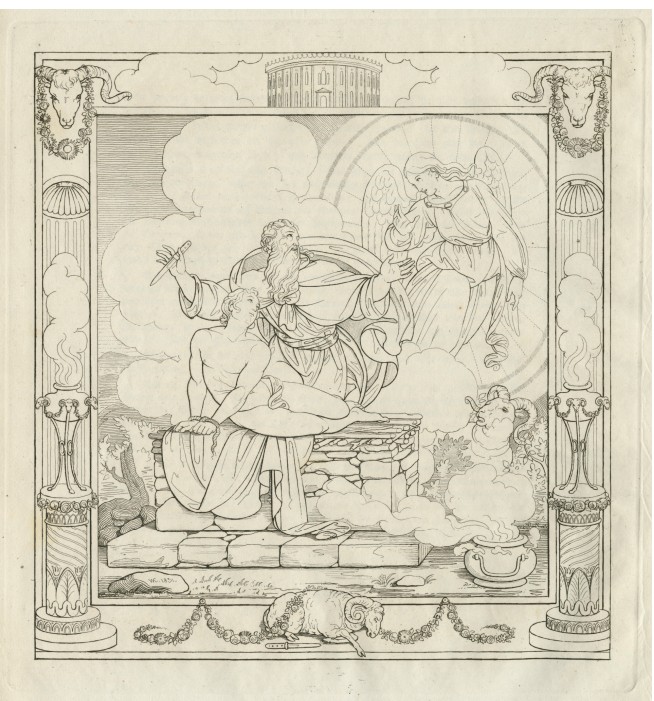

**Figure 5.** Wilhelm Georg Alexander von Kügelgen. *Abraham's Sacrifice*. Part of the Series *Die Geschichte des Reichs Gottes, nach der heiligen Schrift, in Bildern von Wilhelm von Kügelgen.* (Krummacher 1833). Etching. 35 × 27.5 cm. Art Museum of Estonia, Tallinn. EKM j 58798:7 G 30343:7.

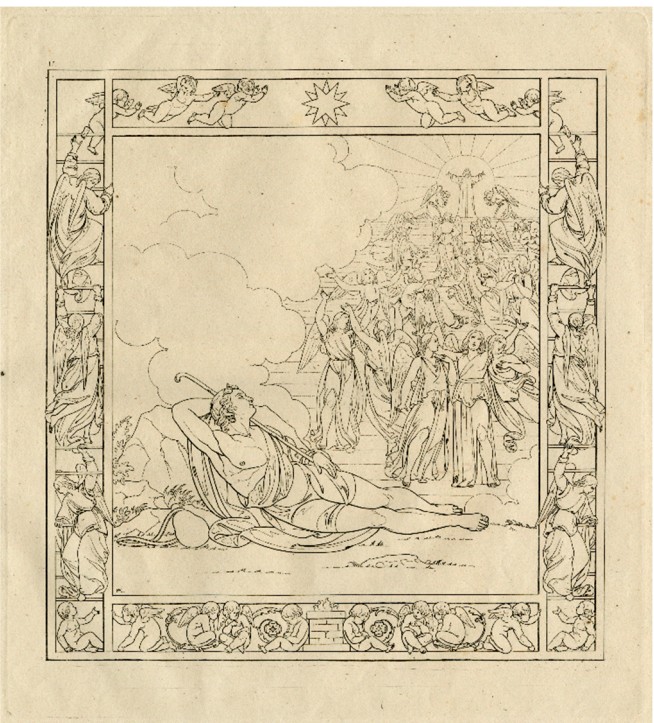

**Figure 6.** Wilhelm Georg Alexander von Kügelgen. *Jacob's Ladder*. Part of the Series *Die Geschichte des Reichs Gottes, nach der heiligen Schrift, in Bildern von Wilhelm von Kügelgen.* (Krummacher 1833) Etching. 35 × 29 cm. Art Museum of Estonia, Tallinn. EKM j 58798:10 G 30343:10.

Hippius also made religious prints based on the works of Raphael, Michelangelo, and Domenico Ghirlandaio, such as lithographs depicting the heads of various saints (Figure 7). An avid copyist was Carl Timoleon Neff, who painted a copy of Raphael's *Madonna della*

*Sedia* (undated). Such works based on that of the Renaissance masters raise questions regarding artists' intentions and their copies as actors. Religious art seeks to depict what is otherwise invisible, such as the Christ, the Saints, and Biblical events. These images are thus important actors in religion, making its content easier to imagine and experience and bringing it closer to the audience. Although the Reformation strove to enable education and promote vernacular translations of the Bible, the Lutheran church also valued the didactic and mnemonic functions of religious art (Kaufmann 2002). Furthermore, religious art made a significant contribution in creating mental images of Biblical scenes and characters. Images of Christ, angels, and demons were largely created through religious art and then reinforced in subsequent artworks (Burke 2001, p. 46). For Hippius, Neff, and other 'Pro-Raphaelites', however, religious content was inseparably tied to the fame of the Renaissance masters. While still depicting religious themes, their copies also mediated the original artworks that remained inaccessible for most of the Baltic audience.

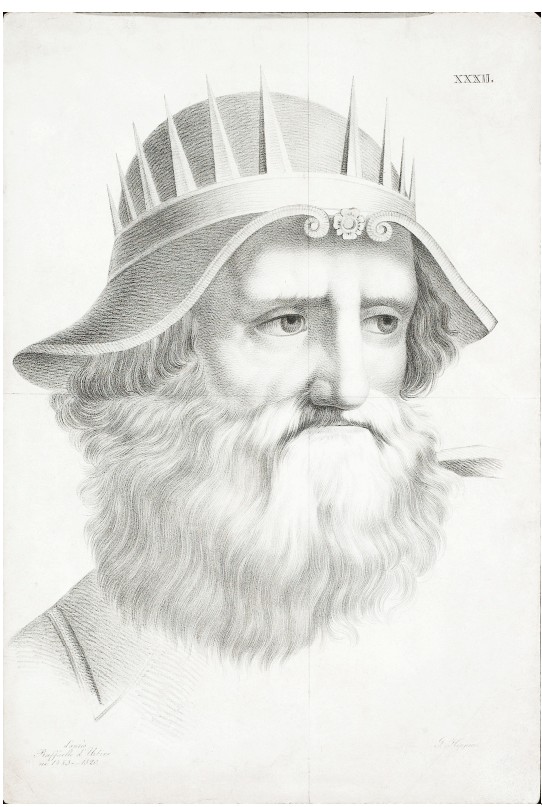

**Figure 7.** Gustav Adolf Hippius. *Head of David*. A Detail from Raphael's Composition *Dispute* in the Vatican. 1830s. Lithograph. 43.1 × 29.4 cm. Art Museum of Estonia, Tallinn. EKM j 153:784/15 G 105.

## 5. Conclusions

Religious art needs to be pictorially specific in order to portray transcendental ideas by sensorial means. How does the Word of the Bible assume a pictorial form? Nazarene artists approached this question in the first half of the nineteenth century by cultivating a generalized and idealized aesthetic approach inspired by Raphael. Biblical characters were depicted in a mild light with harmonious facial features and freely falling robes, often in an Italianesque landscape. This mode of depiction referred to the culture of both Greco-Roman antiquity and the Renaissance art that mediated it, with the objective of reflecting universal values. The Nazarene aesthetic dominated Christian art in the German *Kulturraum* of this period. It sought to re-mystify and aestheticize the physical world and suggest transcendence in the hopes of redirecting contemporary society to the piety of an earlier time. According to the Nazarenes, this was made possible primarily by art and

religion. For them, Raphael became a model on which they based their lifestyles in pursuit of the ideal symbiosis of creativity and religiosity.

Early nineteenth-century religious art in the Baltic Provinces reflected the contemporary Lutheran doctrine and rhetoric and a veneration of great art of the past that entailed transmitting style and tradition through copying. The appearance of Baltic German religious art in the early nineteenth century was the product of a confluence of factors. The classical ideal, the admiration of Raphael, contact with other German artists, and discussions about religious art all played a role. Religious paintings served two primary functions: to communicate religious themes and mediate European art history and Old Masters. These motives were expressed in the given work simultaneously at varying levels of prominence depending on the artist's intention and the subsequent purpose and display of the work.

Print copies of well-known paintings also raise the question of which was more important: the depiction of a Biblical event or the copying of a famous Renaissance master? Copying well-known works was an established technique for presenting a subject in the conservative, tradition-bound field of religious art. The resultant works both represented and recreated the already existing canon of art history and appealed to the values seen in Renaissance art. In the Baltic context, they acted as mediators of the Italian Renaissance and allowed artists to present themselves as part of European culture. The classical ideal was thus perpetuated by a tightly connected network in which Baltic German artists joined others in re-establishing the power of the European art canon.

**Funding:** This research received no external funding.

**Data Availability Statement:** Not applicable.

**Acknowledgments:** This is a translation of *"Prorafaeliidid". Religioosse kunsti klassikaline ideaal ja teoste toimijapotentsiaal 19. sajandi esimesel poolel Eesti- ja Liivimaal* originally published in Estonian by Eesti Kunstiteadlaste Ühing (Estonian Society of Art Historians and Curators) in Kunstiteaduslikke Uurimusi (Studies on Art and Architecture) 2020 29/1–2, pp. 90–120. This translation was prepared by Liisa-Helena Lumberg-Paramonova. Permission was granted by Liisa-Helena Lumberg-Paramonova.

**Conflicts of Interest:** The author declares no conflict of interest.

## Notes

[1]　Among Baltic German artists, only Eduard Karl Franz von Gebhardt (1838–1925, from Järva-Jaani, Estonia) was dedicated to religious works. Gebhardt was influential as a professor in the Düsseldorf Art Academy and considered to be an innovator of religious art in Germany. Foregoing the Nazarene style, he turned to 16th–17th century German and Dutch artists for inspiration. In addition, the theoretically high position of religious art was not always reflected in practice—religious art was not bought or shown more than other genres in Europe (see, for example, Aston 2009, p. 291).

[2]　This work is currently under conservation at Conservation and Digitisation Centre Kanut. The exact identity of baron Clodt remains unconfirmed. Based on the date, it might be Adolf Friedrich Clod(t) von Jürgensburg (1738–1806), a forefather of later artists such as Peter Jakob, Georg Gustav, and Michael Clodt von Jürgensburg, who were active in the St. Petersburg Art Academy. Attributing this work to baron Clodt might also be a mistake as ongoing conservation work shows no signature. This entry in the National Registry of Cultural Monuments names the work *Ascension*, whereas its actual title is Transfiguration (16419 Altar painting *Ascension* n.d.).

[3]　There are many press announcements that confirm this practice (for example, Halliste ue kirriko ehhitussest ja pühhitsussest 1867; E. W. 1850). I also wrote about altar paintings in Estonian Lutheran churches from the second half of the 19th century in my master's thesis (Lumberg 2017).

[4]　I am not aware of the provenance of these portraits; Pezold mentions making these.

[5]　I am grateful to professor Heikki Hanka from Jyväskylä University for this information.

[6]　Franz Gerhard von Kügelgen was born in Bacharach and died near Dresden but spent a major part of his active life in the Baltic Provinces, along with his twin brother, Karl Ferdinand von Kügelgen.

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
