# Peer review of "“Pro-Raphaelites”: The Classical Ideal in Religious Art and the Agency of Artworks in Estonia from 1810 to 1840"

_arts, 2019_

Round 1
Reviewer 1 Report
Such a beautifully written and clear paper. The premise is both engaging and thought provoking. Well done!
Figure 5 is very dark. Can you lighten it?
Author Response
Thank you for your review and kind assessment!
I took a look at Figure 5 and it seemed alright in my computer. Perhaps there was a technical issue, but I could not detect that. Hopefully it will turn out as it should in the end version.
Reviewer 2 Report
Judging from the title, annotation and introduction, the article apparently claims to explore the situation in the Baltic Provinces (e. g. present-day Estonia and Latvia). Nevertheless, the analysis is written exclusively from the perspective of present-day Estonia. No visual or written evidence related to Latvia is used. The famous Ernst Gotthilf Bosse and his 1819 copy of Raphael's Transfiguration as altarpiece for Riga Dom Cathedral are not even mentioned.
The Estonia-related perspective can be preserved, but it should be clearly stated modifying the annotation, the introduction and perhaps even the title of the article.
For getting informed about the research situation in Latvia, the author may start with such reference works and source publications as: 1) chapters by various researchers in: Art History of Latvia. Vol. 3: 1780–1890. Ed. E. Kļaviņš, Books 1–2. Riga: Institute of Art History of the Art Academy of Latvia; Art History Research Support Foundation, 2019; 2) Secundum artem: 19. gadsimta akadēmiskā glezniecība = Secundum artem: 19th Century Academic Painting. [Exh. cat.]. Ed. K. Rudzīte. Rīga: Valsts Mākslas muzejs, 2003; 3) J. Döring, Was ich nicht gern vergessen möchte oder Erinnerungen aus meinem Leben. (Historische Quellen VIII.) Hg. v. V. Kvaskova. Riga: Lettisches Nationalarchiv, 2016.
On lines 128–128 author writes: "For example, Ecce homo, by Guido Reni, one of the ‘stars’ of Christian art, was used by the Baltic German artist Otto Friedrich Theodor von Moeller for the head of Christ on his altar paintings in churches in Harju-Jaani (St. Johannis, 1872) and Võnnu (Wenden, 1873).xvi" The last of the two place-names – Võnnu – should be replaced with the Latvian name of this Latvian town: Cēsis.
For other minor comments, see the attachment.

Author Response
Thank you for your thorough review and helpful comments!
Point 1:
Judging from the title, annotation and introduction, the article apparently claims to explore the situation in the Baltic Provinces (e. g. present-day Estonia and Latvia). Nevertheless, the analysis is written exclusively from the perspective of present-day Estonia. No visual or written evidence related to Latvia is used. The famous Ernst Gotthilf Bosse and his 1819 copy of Raphael's Transfiguration as altarpiece for Riga Dom Cathedral are not even mentioned.
The Estonia-related perspective can be preserved, but it should be clearly stated modifying the annotation, the introduction and perhaps even the title of the article.
Answer:
It is true that there was a discrepancy between the title/claims and the resulting article itself. I have now corrected that, changing some sections in the introduction and also the title. I kept the Estonia-related perspective, but included a reference to Bosse's important altar painting as well - omitting that was definitely not right and happened by my mistake. Thank you for pointing that out. Comparing my previous knowledge/research from Estonia with art history of Latvia should have already been more thorough, but I shall continue working on that.
Point 2:
On lines 128–128 author writes: "For example, Ecce homo, by Guido Reni, one of the ‘stars’ of Christian art, was used by the Baltic German artist Otto Friedrich Theodor von Moeller for the head of Christ on his altar paintings in churches in Harju-Jaani (St. Johannis, 1872) and Võnnu (Wenden, 1873).xvi" The last of the two place-names – Võnnu – should be replaced with the Latvian name of this Latvian town: Cēsis.
Answer:
This has now been corrected.
Point 3:
There was a question in the file regarding Baron Clodt: Do his first names and life dates remain unknown?
Answer:
Unfortunately yes. I have attempted to specify this and added some gained knowledge to the endnote. It might be Adolf Friedrich Clod von Jürgensburg (1738–1806), but I cannot be sure. This painting is now under conservation in Kanut in Tallinn and I will be later able to see it more closely. There is some contradictory evidence regarding the painting.
As to other comments that you had in the file, I have made changes accordingly and now submit the renewed, edited article.
Reviewer 3 Report
This insightful essay covers the Nazarene aesthetic and its influence on Baltic Provinces from 1810 to 1840, especially the Pre-Raphaelite art form. This article has taken steps to understand better how specific biblical characters and allusions are discussed in the essay, yet kept the unschooled reader in mind and provides a brief introduction to the subject, complete with helpful bibliographies, practical illustrations, and a step-by-step demonstration of how Nazarene embody or think through these issues. Not only art students but also the religious-philosophic observers committed to expository historical reading that is hugely relevant to this article.
Below is the assessment of the article:
Introduction: A clear introductory may be better to quick-brief an overview of the discussions following.
Religious art in Estonia: Good starting point for developing the need for religious art in Estonia.
Brotherhood of St Luke and Nazarenes: Frederich Overbeck (1789-1869) The Three Ways of Art may help shed light on the development (Harrison, Wood & Gaiger, Art in Theory: 1648-1815, 2001: 1131-4).
Religious art in practice: It would be good to show any aberrant.
Conclusion: Fair assessment; the author can include article limitations.
This essay fills a gap by providing serious readers of religious art or mystery with a helpful tool for interpreting biblical characters and allusions in contemporary art. The author introduces the Nazarene aesthetic to modern readers in the area of how the Pre-Raphaelites used the biblical transcendental ideas by sensory means, laying out in the rest of the essay to determine further the sort of biblical nuances that have been used or probably misused by the Pre-Raphaelite art practitioners.
The author can further elaborate their methodical approach that will reveal the depth, beauty, interpretative richness, and unity of Scripture and the Nazarene aesthetic, including the beautiful ways in which these uses to help contemporary practitioners and readers understand their relationship to the divine or transcendences and its immediate cultural extension within the context of the unfolding redemptive-historical narratives of Scripture.
Author Response
Thank you for your comments and insights!
Point 1:
Introduction: A clear introductory may be better to quick-brief an overview of the discussions following.
Answer:
I have included a paragraph to the end of the introduction that gives an overview of the following.
Point 2:
Brotherhood of St Luke and Nazarenes: Frederich Overbeck (1789-1869) The Three Ways of Art may help shed light on the development (Harrison, Wood & Gaiger, Art in Theory: 1648-1815, 2001: 1131-4).
The author can further elaborate their methodical approach that will reveal the depth, beauty, interpretative richness, and unity of Scripture and the Nazarene aesthetic, including the beautiful ways in which these uses to help contemporary practitioners and readers understand their relationship to the divine or transcendences and its immediate cultural extension within the context of the unfolding redemptive-historical narratives of Scripture.
Answer:
Thank you for suggesting literature to better understand art theory of the Nazarenes! In this article, however, I did not include more ideas regarding the Nazarenes and their understanding of art as my primary goal was to examine Baltic German art and to focus on local Baltic artworks. The scope of this article did not enable to delve more deeply into Nazarene aesthetic, but I shall keep these comments in mind.
Point 3:
Religious art in practice: It would be good to show any aberrant.
Answer:
This is a good suggestion, but taking into account the scarcity of Baltic German art in Estonia around 1800, there is too little evidence or preserved artworks that would enable to show aberrants. Research this far shows rather the unity in religious art. Portraits and landscapes were much popular genres, leaving religious art to be practiced rather in small numbers.
Point 4:
Conclusion: Fair assessment; the author can include article limitations.
Answer:
Some limitations were mentioned in the added paragraph in the introduction, such as limitations regarding geography (pointed out by Reviewer 2).